# Is it really always only the others who are to blame? GP's view on medical overuse. A questionnaire study

**Maximilian Pausch**[1], **Angela Schedlbauer**[2], **Maren Weiss**[3], **Thomas Kuehlein**[2], **Susann Hueber**[2]*

**1** Faculty of Medicine, Friedrich-Alexander-University Erlangen-Nürnberg (FAU), Erlangen, Germany,
**2** Institute of General Practice, Universitätsklinikum Erlangen, Erlangen, Germany, **3** Institute of Psychology, Friedrich-Alexander-University Erlangen-Nürnberg (FAU), Erlangen, Germany

* susann.hueber@uk-erlangen.de

## Abstract

### Background

Medical overuse is a common problem in health care. Preventing unnecessary medicine is one of the main tasks of General Practice, so called quaternary prevention. We aimed to capture the current opinion of German General Practitioners (GPs) to medical overuse.

### Methods

A quantitative online study was conducted. The questionnaire was developed based on a qualitative study and literature search. GPs were asked to estimate prevalence of medical overuse as well as to evaluate drivers and solutions of medical overuse. GPs in Bavaria were recruited via email (750 addresses). A descriptive data analysis was performed. Additionally the association between doctors' attitudes and (1) demographic variables and (2) interest in campaigns against medical overuse was assessed.

### Results

Response rate was 18%. The mean age was 54 years, 79% were male and 68% have worked as GP longer than 15 years. Around 38% of medical services were considered as medical overuse and nearly half of the GPs (47%) judged medical overuse to be the more important problem than medical underuse. Main drivers were seen in "patients´ expectations" (76%), "lack of a primary care system" (61%) and "defensive medicine" (53%), whereas "disregard of evidence/guidelines" (15%) and "economic pressure on the side of the doctor" (13%) were not weighted as important causes. Demographic variables did not have an important impact on GPs´ response pattern. GPs interested in campaigns like "Choosing Wisely" showed a higher awareness for medical overuse, although these campaigns were only known by 50% of the respondents.

**Data Availability Statement:** All relevant data are within the paper and its Supporting Information files.

**Funding:** The authors received no specific funding for this work.

**Competing interests:** The authors have declared that no competing interests exist.

## Discussion

Medical overuse is an important issue for GPs. Main drivers were searched and found outside their own sphere of responsibility. Campaigns as "Choosing Wisely" seem to have a positive effect on GPs attitude, but knowledge is still limited.

## Introduction

Medical overuse is a common problem in health care [1, 2]. Despite frequent discussions in public, politics and the medical community, there is still no consistent concept for defining and measuring medical overuse [3, 4]. Medical overuse is often described as "a health care service [that] is provided under circumstances in which its potential for harm exceeds the possible benefit" [5]. This rather simplified definition only corresponds to obviously ineffective services. However, many services are in a "nebulous grey zone, where evidence is lacking or weak." [2]. In this grey zone, patients' and physicians' attitudes and beliefs might play the ultimate role in determining and tackling with low value care [6].

Much of the current literature pays particular attention to drivers and solutions of medical overuse [see for example 7, 8, 9]. Factors promoting medical overuse in general practice were attributed to internal factors such as physicians' need for reassurance and the belief that action is better than inaction [10]. Also, cognitive biases in medical decision making and an insufficient ability of dealing with uncertainty seem to play an important role [11, 12]. New medical technology and its general availability are frequently misleading physicians to use it whether appropriate or not [13]. Causes of medical overuse were also attributed to external factors such as patient expectations [14] and fear of litigation resulting in defensive medicine [15].

Estimations of the extent of medical overuse range from 10% to 30% of the total expenditure in the US healthcare system depending on the respective definition and research method [16, 17]. In 2011, between $158 and $226 billion were spent for overtreatment [17–19]. Beside the economic burden, there is an important medical impact. Unnecessary investigation and treatment will frequently result in psychological and physical harm [20].

In light of lowering risk-factor thresholds and disease mongering the prevention of medical overuse–also named quaternary prevention—is getting more and more important [21]. Supporting physicians in this task is the important purpose of campaigns like "Choosing Wisely"[22]. Previous studies have shown that "Choosing Wisely" can have a positive impact on physicians' attitudes, but are still not known enough by the vast majority of practicing doctors [23, 24].

The GP's role as a gate-keeper is weakened considerably in the German health care system, where patients have direct access to specialists without the need to be referred. Nevertheless, GPs might be in a central position for preventing medical overuse, considering that it is easier not to initiate a cascade of diagnostic tests and their therapeutic consequences, rather than trying to stop it when it is in the full run [25, 26].

In a previous study, we qualitatively explored perceptions and opinions of medical overuse amongst German GPs [27]. They perceived medical overuse as a common problem in the German health care system, mainly caused by drivers like defensive medicine, lack of a primary care system and patients' expectations. Solutions proposed were reducing defensive medicine, focusing on shared-decision-making and conducting stepwise diagnostic investigation. In this current study, we wanted to quantify these data. The primary aim was to capture the current opinion of GPs regarding medical overuse. The secondary aim was to assess the association

between doctors' attitudes and (1) demographic variables, and (2) campaigns´ awareness and interest.

## Materials and methods

### Study design and sample

We conducted an online questionnaire study. The study is reported following the STROBE and CHERRIES statement [28, 29]. The questionnaire was developed following the generalization model described by Mayring [30]. In order to gather general conclusions about a specific topic, the results of a qualitative study can be used to develop a quantitative study.

Referring to the results of the qualitative study by Alber et al [31], we decided to consider the following domains of medical overuse as to be important: perceived relevance, drivers of medical overuse and solutions. In the development of the items, both findings of a literature search [4, 16, 32–38] and experts' opinion were included. The software SurveyMonkey was used for programming the questionnaire [39].

A pre-test of the questionnaire evaluating its design and content was performed with nine physicians and two medical students. Pre-tests were carried out using the "thinking aloud" method [40]. For the recruiting of participants, the following inclusion criteria were applied: Qualified GPs and GP trainees, specialist doctors in internal medicine or physicians without specialist training working as primary care physicians.

The link to the questionnaire was sent via email together with an invitation to participate. Email addresses were selected from the registry of GPs in the Bavarian Association of Statutory Health Insurance Physicians, short Bavarian ASIP. (The Bavarian ASIP is one of the healthcare system's self-administration bodies, responsible for ensuring that outpatient medical treatment is provided and for the distribution of payments to physicians in ambulatory care). In the case of missing email addresses in the ASIP registry, addresses were searched via Internet (Google search) resulting in a total of 750 addresses.

An email containing a link to the survey, a description of study's objective, information on data handling (anonymity), the informed consent statement and the invitation to participate was delivered to GPs in North and East Bavaria on 20 June 2017. A reminder was sent one week later. Data collection was closed 18 days after the first mail.

Ethical approval was granted by the Ethics Committee of the Faculty of Medicine of the Friedrich-Alexander University Erlangen-Nürnberg (91_17 B, 07.04.2017).

### Measures

The questionnaire consisted of two parts. Firstly, we collected demographic variables such as gender, age, professional title and experience, area of work and practice volume. Secondly, we asked 36 questions on the following topics in the main section of our questionnaire:

 **A. General assessment of medical overuse.**

1. *Estimate of the prevalence of medical overuse in health services*: "How high would you estimate the percentage of medical overuse in Germany at the moment?" on a scale between 0 and 100%.

2. *Perceived need for action*: "Where do you currently see more need for action regarding the quality of treatment for our patients?" Decision has to be made on a visual analogue scale between medical underuse and medical overuse.

3. *Perceived causes for medical overuse*: Out of a list of eight suspected causes such as patient expectation, marketing of the pharmaceutical industry, disease mongering, the three

most important sources of medical overuse should be selected. There was no ranking option, just a selection mode.

4. *Campaigns' familiarity and use*: selection of "Choosing wisely" [41], "Smarter medicine" [42], "Less is more" [43], "Klug entscheiden" [44], "Quaternary prevention" [25] and "None of the named campaigns".

   **B. Personal attitude towards medical overuse.** Participants were asked to make a decision on a six-point Likert scale. Endpoints ranged from "I completely disagree" to "I totally agree" (topic five and eight) or from"not correct at all" to "fully correct"(topic six and seven).

5. *Tendency to justify medical overuse*: 13 items on medical decision making in everyday practice, e.g., "Patients with acute low back pain seem to be dissatisfied when symptoms are not clarified by imaging techniques."

6. *Perceived relevance of medical overuse*: Six items, e.g., "I know patients being harmed by medical overuse."

7. *Evaluation of approaches to prevent medical overuse*: Six items, e.g., "As a doctor, one should talk to one's patients about the costs of tests and medication".

8. *Evaluation of "Choosing Wisely" advices for family physicians*: Six items, e.g., "Don't perform imaging for low back pain within the first six weeks unless red flags are present."

*Reasons for the questionnaire design chosen*: GPs were asked to estimate the prevalence of medical overuse at the beginning of the survey (rather than at the end) in order to avoid bias caused by confronting them with the subject of medical overuse throughout answering the questionnaire. A Likert scale with an even number of categories was chosen so that our respondents had no neutral category and were forced to pick an option for or against an item [45]. Items were randomly assigned for each participant in the topics to prevent hidden biases due to the same item sequence presentation (order effect bias) [46]. In question three, participants had to select three of the most substantial causes suspected for medical overuse–so called forced choice question [45]. It was not possible to flip backwards in the questionnaire and modify items retrospectively in order to prevent social desirability bias [47]. GPs should not be able to modify answers based on an upcoming feeling that the authors have a preference which they want to hear. The full questionnaire is available as supplementary file.

## Data analysis

Only completely answered questionnaires were included in the analysis. Statistical analyses were performed using SPSS, version 24 (IBM Statistics). The processing of the data was carried out in two steps: A descriptive analysis followed by an evaluation of group differences. For all items on the six-point Likert scale mean and standard deviation were computed. A box plot was chosen for graphical presentation.

Participants were grouped according to their *demographic variables* including gender (male vs. female), age ($\leq$ 50 years vs. > 50 years) and practice volume (patients treated per physician per quarter ($\leq$ 1000 vs. > 1000). In addition, respondents were grouped according to *being interested or not being interested in the mentioned campaigns* resulting in two groups: "Campaign interested GPs", who had already heard at least of one campaign and "non-interested GPs", who had heard of none. A similar approach was chosen by Kost et al. [24].

Differences between groups were assessed in regard to the main topics of the questionnaire: "*General assessment of medical overuse*" (section A) and "*Personal attitudes towards medical overuse*" (section B). For section B, a sum score for each of the four subtopics was built. A sum

score is defined as the sum of all numerical item values per participant summed up in one topic of the questionnaire [42]. For a reasonable and homogeneous interpretation of the sum score, all items must have the same item polarization [43]. Therefore, we exchanged the scale endpoints of the items 5.1, 5.12, 6.1, 6.3 and 6.4. Cronbachs Alpha was used to test whether a sum score for each subtopic was reasonable (see below). Cronbachs Alpha is low for two subtopics, which in part might be due to the low number of items per subtopic. A higher sum score indicated:

- higher tendency to justify medical overuse (topic 5): Cronbachs $\alpha$ = 0.64, sum score range (R) = 13–78,

- higher rating of relevance and frequency (topic 6): Cronbachs $\alpha$ = 0.55, sum score range (R) = 6–36

- higher acceptance of proposed solutions for preventing medical overuse (topic 7): Cronbachs $\alpha$ = 0.39, sum score range (R) = 6–36

- higher acceptance of Choosing wisely advices (topic 8): Cronbachs $\alpha$ = 0.71, sum score range (R) = 6–36

Groups were compared using Students t-Tests.

## Results

### Descriptive analysis

*Response rate*: We invited 750 practices via email to participate and received 155 (21%) questionnaires of which 135 (87%) were fully completed, resulting in a response rate of 18% that could be used for data analysis.

The mean age of participants was 54 years (SD = 8), 79% were male and 92% were trained GPs. The majority (68%) had professional experience as a GP for over 15 years. Approximately half of the participants regarded their doctor's office to be located in an urban area (51%) and almost two thirds (65%) had a practice volume of more than 1.000 patients per quarter. *Demographic data* can also be seen in Table 1.

**Table 1. Demographic data of respondents.**

|  |  | N | % |
|---|---|---|---|
| Gender | Male | 106 | 79% |
|  | Female | 29 | 22% |
| Age | $\leq$ 50 years | 44 | 33% |
|  | > 50 years | 91 | 67% |
| Professional title | GP | 124 | 92% |
|  | Specialist for internal medicine (working as GP) | 9 | 7% |
|  | Non-specialist medical doctor | 1 | 1% |
|  | GP trainee | 1 | 1% |
| Professional experience | $\leq$ 15 years | 43 | 32% |
|  | > 15 years | 92 | 68% |
| Area of work | Urban | 69 | 51% |
|  | Rural | 66 | 49% |
| Practice volume | $\leq$ 1.000 | 47 | 35% |
|  | > 1.000 | 88 | 65% |

**Table 2. Frequency of presumed causes for medical overuse ().**

|  | Relative frequency |
|---|---|
| 1. Patients´ expectations | 76% |
| 2. Lack of primary care system | 61% |
| 3. Defensive medicine | 53% |
| 4. Disease mongering | 34% |
| 5. Marketing of the pharmaceutical industry | 27% |
| 6. Progress in medical technology | 22% |
| 7. Disregard of evidence/guidelines | 15% |
| 8. Economic pressure on the side of the doctor | 13% |

Information on the demographics about the entire population of GPs in Northern Bavaria are provided by the Bavarian Association of Statutory Health Insurance Physicians (Bavarian ASIP, described above). In the entire population, mean age is 55.3 years and 64% are male. That is, our sample was only slightly younger and had higher percentage of men.

**A. General assessment of medical overuse.** *Frequency estimation of medical overuse*: Around 38% of medical services were considered as overuse.

*Perceived need for action*: *Medical underuse or overuse*: Nearly half (47%) of the respondents judged medical overuse to be the more important problem. For 28% it was more important to tackle medical underuse.

*Presumed causes for medical overuse*: Factors contemplated most frequently were "patients´ expectations" (76%), "lack of a primary care system" (61%), and "defensive medicine" (53%), whereas "disregard of evidence/guidelines" (15%) and "economic pressure on the side of the doctor" (13%) appeared least frequently. Results can also be seen in Table 2.

*Campaign awareness and interest*: Half of the participants (50%) had never heard of the campaigns. "Choosing wisely" was known to 32%, of whom 88% stated that they had also used it. The German offshoot "Klug entscheiden" was known to 30%, of which 80% had already taken a closer look at the respective items. Results can be seen in Table 3.

**B. Personal attitude towards medical overuse.** Selected results are shown below. All results are depicted in Figs 1 to 3.

*Tendency to justify medical overuse*: Most of the doctors perceived that patients with unspecific back pain seemed dissatisfied if their symptoms were not being checked via diagnostic imaging (Item 5.4, $M = 4.1$, $SD = 1.4$). The majority agreed that patients associated competence

**Table 3. Campaign awareness and interest.**

|  | Campaign interested GPs, that is the proportion of all GPs that are being aware of a campaign | The proportion of those also being familiar with the campaign or who had used at least elements of a certain campaign |
|---|---|---|
| Choosing wisely | 32% | 88% |
| Smarter medicine | 4% | 17% |
| Less is more | 19% | 44% |
| Klug entscheiden | 30% | 80% |
| Quaternary prevention | 18% | 50% |
| None of these campaigns | 50% | / |

| Tendency to justify medical overuse | | MW (SD) | Distribution of scores |
|---|---|---|---|
| | | | **1** = *I completely disagree* — *I totally agree* = **6** |
| 5.1 | I consider individual health services (IGeL) to be a form of overtreatment. | 3.6 (1.8) | |
| 5.2 | I consider extensive laboratory tests during check-ups to be useful in order to filter individual cases. | 3.4 (1.6) | |
| 5.3 | Visits by pharmaceutical representatives can be helpful in order to be informed over new product developments. | 3.1 (1.6) | |
| 5.4 | Patients with unspecific back pain are dissatisfied if their symptoms are not being checked via imaging processes. | 4.1 (1.4) | |
| 5.5 | A GP is attributed with more competence if he/she conducts more diagnostic tests. | 3.8 (1.3) | |
| 5.6 | Patients are more easily satisfied through actions and the readiness to act than through a "wait and see" approach. | 4.2 (1.4) | |
| 5.7 | I would rather overuse forms of treatments once than to miss something once. | 3.7 (1.3) | |
| 5.8 | Liability processes in the medical field lead to overuse since as a result doctors want to protect themselves in regard to the diagnosis. | 4.7 (1.3) | |
| 5.9 | Assessment portals lead to medical overuse since as a result patients consider themselves more and more to be consumers. | 4.2 (1.5) | |
| 5.10 | Temporal expenditure is often the reason for the failure of appropriate information about benefits and harm of diagnosis and therapy. | 4.0 (1.5) | |
| 5.11 | Guidelines are an infringement of professional liberties of doctors in order to reduce costs. | 2.5 (1.5) | |
| 5.12 | I have already consciously decided against therapeutic measures, even though the newly implemented lower threshold would have suggested treatment of some form. | 4.5 (1.6) | |
| 5.13 | I want to figure out the cause for my patient's symptoms as quickly as possible. | 4.7 (1.1) | |

**Fig 1. Descriptive analysis of items on personal attitudes to medical overuse (Item 5.1 to Item 5.13).** For each item, mean value and standard deviation are presented. A box plot for each item is shown in the last column. It consists of the minimum and maximum, the interquartile range and the median. The black dot represents the mean.

with performing more diagnostic tests (Item 5.5, *M* = 3.8, *SD* = 1.3) and that they want to figure out the causes for symptoms as soon as possible (Item 5.13, *M* = 4.7, *SD* = 1.1.). A high proportion admit that defensive medicine leads to medical overuse (Item 5.8, *M* = 4.7, *SD* = 1.3). A majority of GPs agreed to the statement that they already have "decided against

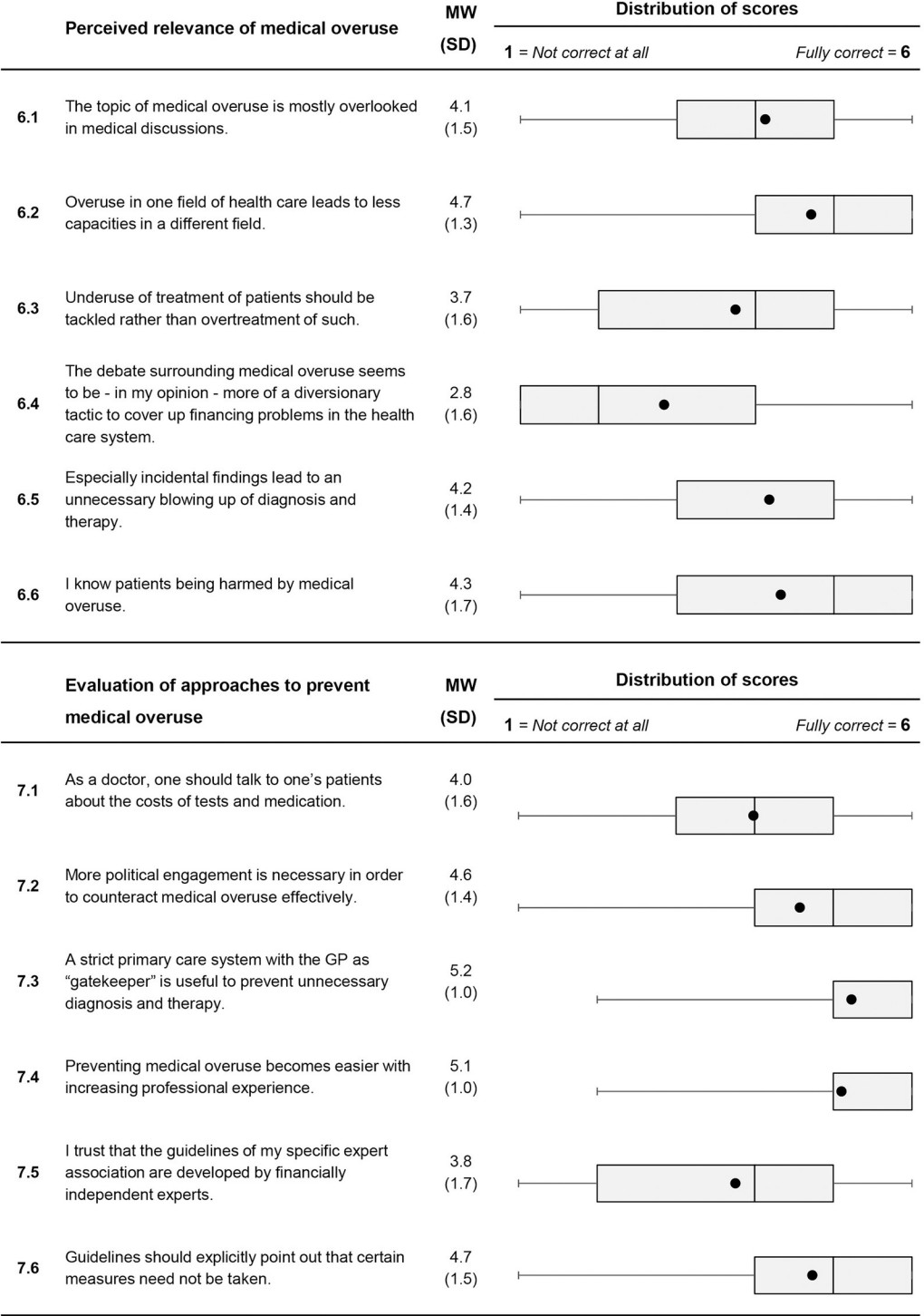

**Fig 2. Descriptive analysis of items on personal attitudes to medical overuse (Item 6.1 to Item 6.6 and Item 7.1 to Item 7.6).** For each item, mean value and standard deviation are presented. A box plot for each item is shown in the last column. It consists of the minimum and maximum, the interquartile range and the median. The black dot represents the mean.

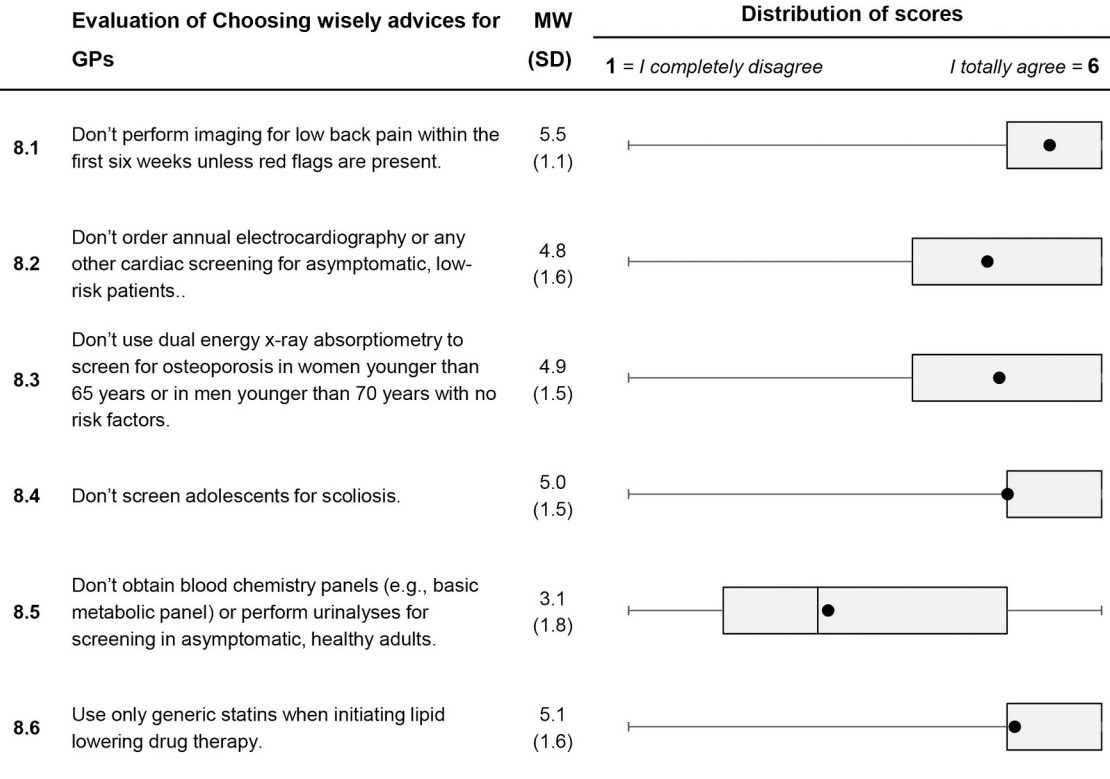

**Fig 3. Descriptive analysis of items on personal attitudes to medical overuse (Item 8.1 to Item 8.6).** For each item, mean value and standard deviation are presented. A box plot for each item is shown in the last column. It consists of the minimum and maximum, the interquartile range and the median. The black dot represents the mean.

therapeutic measures, even though the newly implemented lower threshold would have suggested treatment of some form" (Item 5.12, $M = 4.5$; $SD = 1.6$).

*Perceived relevance of medical overuse*: A majority was aware of incidental findings and related problems (Item 6.5, $M = 4.2$; $SD = 1.4$). Doctors mostly agreed to the statement of knowing patients being harmed by medical overuse (Item 6.6, $M = 4.3$; $SD = 1.7$). At the same time, a high proportion acknowledged that "The topic of medical overuse is mostly overlooked in medical discussions." (Item 6.1, $M = 4.1$; $SD = 1.5$).

*Evaluation of approaches to prevent medical overuse*: GPs supported further action of politics by mostly agreeing with the item "More political engagement is necessary in order to counteract medical overuse effectively."(Item 7.2, $M = 4.6$; $SD = 1.4$). The gatekeeper role was considered important to almost all respondents (Item 7.3, $M = 5.2$; $SD = 1.0$). Finally, a high proportion emphasized the role of long professional experience in avoiding medical overuse (Item 7.4; $M = 5.1$; $SD = 1.0$).

*Evaluation of "Choosing Wisely" advices for family physicians*: Recommendations were well acknowledged as illustrated by high agreement rates for most of the recommendations. However, fewer doctors agreed to the recommendation "Don't obtain blood chemistry panels or perform urine analyses for screening in asymptomatic, healthy adults" (Item 8.5; $M = 3.1$; $SD = 1.8$).

## Group differences

*Demographic variables*: Most demographic variables did not show a significant influence on response patterns. Only a higher number of younger doctors ($\leq$ 50 years) agreed to

**Table 4. Results of general assessment of medical overuse and personal attitudes towards medical overuse of groups differing in gender, age and practice volume.**

| | Gender | | Age | | Practice volume | |
|---|---|---|---|---|---|---|
| | **Male** | **Female** | **≤ 50 years** | **> 50 years** | **≤ 1000 patients** | **> 1000 patients** |
| Number (%) | 106 (79%) | 29 (22%) | 47 (35%) | 88 (65%) | 44 (33%) | 91 (67%) |
| **A. General assessment of medical overuse** | | | | | | |
| Frequency estimation of medical overuse M (%): | 38 | 36 | 39 | 37 | 40 | 37 |
| Need for action: | | | | | | |
| 1. Medical overuse | 47% | 45% | 51% | 44% | 39% | 51% |
| 2. Medical underuse | 27% | 38% | 28% | 28% | 39% | 23% |
| **B. Personal attitude towards medical overuse** | | | | | | |
| Tendency to justify medical overuse (sum score, range 13–78) | 48 | 50 | 47 | 49 | 47 | 49 |
| Relevance of medical overuse (sum score, range 6–36) | 23 | 23 | 24 | 23 | 24 | 23 |
| Approaches to prevent medical overuse (sum score, range 6–36) | 27 | 26 | 28* | 26* | 28 | 27 |
| "Choosing Wisely" advices (sum score, range 6–36) | 29 | 27 | 30 | 28 | 24 | 23 |

approaches to prevent medical overuse ($t = 2.145$, $CI = [0.129; 3.191]$, $p = 0.034$). Results are also shown in Table 4.

*Campaign awareness and interest*: GPs interested in campaigns rated the necessity to avoid medical overuse higher than non-interested GPs (campaign interested GPs = 54%, non-interested GPs = 40%, $p = 0.023$). Non-interested GPs showed a higher tendency to justify medical overuse as compared to campaign interested GPs (sum score: non-interested GPs = 50 vs. campaign interested GPs = 47; $p = 0.033$). Campaign interested GPs showed a higher awareness of the relevance of medical overuse (sum score: non-interested GPs = 21, campaign interested GPs = 25, $p = 0.001$) and approval of "Choosing Wisely" recommendations (sum score: non-interested GPs = 26 vs. campaign interested GPs = 30, $p = 0.001$). Both groups rated patients´ expectations, lack of a primary care system and defensive medicine as most influential causes for medical overuse. The frequency of medical procedures considered as overuse was estimated only slightly higher by campaign interested GPs (campaign interested = 39% vs. non-interested GPs = 36%, $p = 0.353$). Results are depicted in Table 5.

## Discussion

Medical overuse was seen as a relevant problem that needs to be tackled, even though GPs agreed that overuse is barely discussed in the medical community. Also, knowledge of

**Table 5. Results of general assessment of medical overuse and personal attitudes towards medical overuse of campaign-interested GPs vs. non-interested GPs.**

| | Non-interested GPs | Campaign interested GPs | |
|---|---|---|---|
| **A. General assessment of medical overuse** | | | |
| Amount (%) | 68 (50%) | 67 (50%) | |
| Relative frequency estimation of medical overuse M (%) | 36 | 39 | $t = 0.931$, CI = [-9.204; 3.311], $p = 0.353$ |
| Need for action: | | | |
| 1. Medical overuse | 40% | 54% | $t = 2.307$, CI = [-16.636; -1.279], $p = 0.023$ |
| 2. Medical underuse | 31% | 25% | |
| **B. Personal attitude towards medical overuse** | | | |
| Tendency to justify medical overuse (sum score, range 13–78) | 50 | 47 | $t = 2.154$; $CI = [0.243; 5.703]$, $p = 0.033$ |
| Relevance of medical overuse (sum score, range 6–36) | 21 | 25 | $t = 4.508$; $CI = [-5.444; -2.123]$, $p = 0.001$ |
| Approaches to prevent medical overuse (sum score, range 6–36) | 27 | 27 | $t = 0.226$; $CI = [-1.626;1.293]$, $p = 0.822$ |
| "Choosing Wisely" advices (sum score, range 6–36) | 26 | 30 | $t = 4.09$; $CI = [-5.66; -1.966]$, $p = 0.001$ |

campaigns addressing medical overuse was limited. Nonetheless, recommendations given by the "Choosing Wisely" campaign were well accepted. Main drivers were seen in patients' expectations, lack of a primary care system and defensive medicine. GPs did not rate the disregard of evidence by physicians as a main driver. Instead they accused a pressure to act by patients not accepting a "wait and see" approach. Active medical performing seems also to be attributed with more competence than non-acting. GPs agreed that incidental findings and widening of disease boundaries are a serious problem. Asked about possible solutions, GPs considered it a main political task to create conditions that help to prevent medical overuse. An important answer to tackle medical overuse was considered in the implementation of a primary care system.

Similar to findings from other countries, GPs in Germany regarded medical overuse as an important issue [35, 36]. Their conviction that patients are more satisfied and that doctors are considered as more competent through actions is well known as commission bias in medical decision making [48]. A study with patients suffering from chronic pain revealed that physicians' refusal to prescribe opioids was attributed to distrust or lack of caring by some patients [49]. It seems possible that GPs are afraid that "watchful waiting" might also be attributed in a similar way and as a consequence interferes with the patient-doctor relationship. This fear together with fulfilling perceived patients' expectations might lead to unnecessary medical procedures.

Consistent with other studies, main drivers of overuse were seen in factors outside GPs own responsibility [35, 36, 50]. It is known that attributing causes for failures or negative events to others helps in self-serving but is likely to prevent self-responsibility and behavior change [51]. It seems that medical overuse is mainly discussed by politics, health insurances and academic institutions, rather than by practicing physicians. The results of our study point to the importance of deeply involving practicing physicians in the discussion and in the process of defining causes and solutions of medical overuse. More effort will be needed to strengthen self-responsibility and commitment of GPs in a way that they feel confident to contribute to this issue.

The limited awareness of GPs for campaigns against medical overuse is not unusual [50, 52]. In our study, campaign interested GPs represented a more critical point of view with higher awareness of medical overuse. Asked about agreement with the recommendations of the "Choosing Wisely" campaigns, GPs mostly agreed with them. In addition, GPs agreed that guidelines should explicitly point out not to perform certain therapeutic or diagnostic services. Still it is somewhat surprising that the recommendation not to screen asymptomatic healthy individuals with blood or urine analyses was rejected. We assume that health checks are not only used to screen otherwise asymptomatic adults but were also seen as a possibility to check for medically unexplained symptoms. As long as nothing is found—which is usually the case— the patient is at ease. It seems that the contents of the campaigns are widely accepted in general, but need more and effective promotion to be implemented in everyday practice. As the perception of medical overuse was not affected by demographic variables, it seems that the attitude towards it is rather a question of personality than a question of age or gender. We might have to put more effort in identifying those early adopters in order to support them.

*Strengths and limitations*: Our study represents a first comprehensive empirical investigation to describe GPs' views on medical overuse in Germany. The questionnaire was developed carefully, systematically and following the results of a qualitative study [27]. Response rate of 18% was limited in comparison to other surveys [36, 52]. Reasons might be the lack of incentive and/or a short recruitment time. Demographic variables were relatively close to the real distribution in the respective regions. GPs in our sample were slightly younger and proportion of men was higher. Nonetheless, findings should be interpreted with caution as generalization might be reduced. The voluntary participation and lack of financial compensation for

participation can lead to a potential bias as it can be assumed that mostly GPs who are interested in medical overuse might have been more likely to respond to our questionnaire, skewing the sample. The study design did not allow observing the real behavior of the participants, but enables us to draw conclusions from attitudes to behavior.

## Conclusion

German GPs perceive that medical overuse is a problem that needs to be solved. However, causes and solutions were mainly seen outside their own responsibility and reach. Our findings lead to the conclusion that GPs' own contributions to medical overuse are neglected by them, maybe unconsciously. More effort is needed to increase awareness for medical overuse. Increasing awareness amongst medical students during medical education might be an important step forward. Also, greater efforts are needed to enhance self-efficacy and ownership regarding medical overuse.

## Supporting information

**S1 File.**
(DOCX)

**S2 File.**
(DOCX)

**S1 Data.**
(SAV)

## Acknowledgments

The present work was performed in fulfillment of the requirements for obtaining the degree „Dr. med."for Maximilian Pausch. The authors thank all participating GPs for supporting our study. We also thank our colleagues Anja Deinzer, Nikoletta Lippert and Marie Kluge for their help in the development of the questionnaire, interpretation of the results and the translation of the manuscript and the questionnaire.

## Author Contributions

**Conceptualization:** Maximilian Pausch, Angela Schedlbauer, Susann Hueber.

**Data curation:** Maximilian Pausch, Susann Hueber.

**Formal analysis:** Maximilian Pausch, Susann Hueber.

**Methodology:** Maximilian Pausch, Maren Weiss, Susann Hueber.

**Software:** Maximilian Pausch.

**Supervision:** Thomas Kuehlein.

**Writing – original draft:** Maximilian Pausch.

**Writing – review & editing:** Angela Schedlbauer, Maren Weiss, Thomas Kuehlein, Susann Hueber.

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
