## [Decision Letter · Decision Letter 0]

15 Oct 2019

PONE-D-19-23067

Is it really always only the others who are to blame? GP’s view on medical overuse.

A questionnaire study.

PLOS ONE

Dear Mrs. Hueber,

Thank you for submitting your manuscript to PLOS ONE. After careful consideration, we feel that it has some merit but does not fully meet PLOS ONE’s publication criteria as it currently stands. Therefore, we invite you to submit a revised version of the manuscript that addresses the points raised during the review process. Specifically, the reviewers have concerns about some methodological and analytic issues as well as limitations of the study which have not been fully discussed.

We would appreciate receiving your revised manuscript by Nov 29 2019 11:59PM. To enhance the reproducibility of your results, we recommend that if applicable you deposit your laboratory protocols in protocols.io, where a protocol can be assigned its own identifier (DOI) such that it can be cited independently in the future. For instructions see: http://journals.plos.org/plosone/s/submission-guidelines#loc-laboratory-protocols

Please note while forming your response, if your article is accepted, you may have the opportunity to make the peer review history publicly available. The record will include editor decision letters (with reviews) and your responses to reviewer comments. If eligible, we will contact you to opt in or out. Please note that request to resubmit does not guarantee acceptance of the revised manuscript.

We look forward to receiving your revised manuscript.

Kind regards,

Monika R. Asnani, DM, PhD

Academic Editor

PLOS ONE

Journal Requirements:

1. Please also include a copy of the questionnaire  in the original language as Supporting Information.

Reviewers' comments:

Reviewer's Responses to Questions

**Comments to the Author**

1. Is the manuscript technically sound, and do the data support the conclusions?

Reviewer #1: Yes

Reviewer #2: Partly

2. Has the statistical analysis been performed appropriately and rigorously? 

Reviewer #1: Yes

Reviewer #2: No

3. Have the authors made all data underlying the findings in their manuscript fully available?

Reviewer #1: Yes

Reviewer #2: No

4. Is the manuscript presented in an intelligible fashion and written in standard English?

Reviewer #1: Yes

Reviewer #2: Yes

5. Review Comments to the Author

Reviewer #1: Table 3 - Title of third column could be more specific to show difference from column 2. e.g. proportion of those who had used elements of the campaign

Lines 248-250 - Sentence needs to express idea more clearly

Limitations should include a comment on the low response rate; why this might be so e.g. response rates in electronic surveys by other authors; how this affects study; how this response rate might be improved in subsequent studies

Reviewer #2: The article is on an important topic area—overuse of medical services.

The article is written clearly, and the tables are well designed.

Overall, the authors followed a reasonable study design was followed. They used sound methods for developing and testing their questionnaire. The analysis has two deficits that can be amended before publication.

The used a registry of general practitioners (GPs) for sample selection. I was curious about why the authors limited their sample to GPs, as they noted that in Germany, unlike other countries, GPs do not serve as gatekeepers for access to medical specialists.

The study has two substantial methodological issues that limit the utility of the results. The authors followed sound survey procedures, with the exception of survey fielding. They only allowed 18 days for GPs to return the survey and sent out a single reminder. A longer data collection period and more follow-up probably would have improved the response rate.

Selection Bias. The study had a good sample plan, but only 18% of subjects returned complete questionnaires. An 18% response rate almost certainly entails some sort of selection bias. The authors mention that only respondents who are interested in the topic of medical overuse may have responded to the survey, although there is almost no way to test that. The authors report demographics for the physicians who returned the survey. I don’t know if they have access to demographics for the entire population of physicians in the geographic area, but if so, it would be helpful to compare the demographics of those who returned the survey with the entire population. While demographics would not tell much of a story about self-selection, it would be beneficial to know if there was selection by age or some other characteristic. In particular, since the authors found that younger doctors viewed medical overuse as more of a problem than older doctors, information on age of the sample compared with age of the population of GPs would have bolstered results.

Inefficient use of data. The questionnaire was comprised mainly of Likert-style questions on a scale of 1 to 6. The authors dichotomized responses into positive and negative. They provide no explanation about why they chose to do this, with the result that they lost information.

Suggestions for Revision

1. Compare the mean ages or age quartiles of survey respondents with the age of all GPs

2. Reanalyze the data without dichotomizing the Likert scales

6. PLOS authors have the option to publish the peer review history of their article (what does this mean?). If published, this will include your full peer review and any attached files.

Reviewer #1: No

Reviewer #2: No

---

## [Author Response · Author response to Decision Letter 0]

27 Nov 2019

Point-to-point-reply

PONE-D-19-23067

"Is it really always only the others who are to blame? GP’s view on medical overuse.

A questionnaire study.”

Dear Monika Asnani, 

Please find below our response to reviewers’ comments. We would like to thank the two reviewers for their help in improving this manuscript. We have endeavoured to incorporate the recommendations in the revised version. Please find a detailed description of our revisions below. Line numbers refer to the document: 'Revised Manuscript with Track Changes'.

Reviewers' comments:

Reviewer's Responses to Questions

Comments to the Author

1. Is the manuscript technically sound, and do the data support the conclusions?

Reviewer #1: Yes

Reviewer #2: Partly

OUR RESPONSE: As will be described below, we have added additional information on distribution of age and sex of all GPs in the respective area (line 2017-220) and added a comment on the low response rates (line 367-375). We hope this will help to improve understanding and enable scientific conclusion. 

2. Has the statistical analysis been performed appropriately and rigorously? 

Reviewer #1: Yes

Reviewer #2: No

OUR RESPONSE: As will be described below, data were reanalysed without dichotomizing the Likert Scales. We have decided to present mean values, standard deviations and in order to enable an overview of the distribution of the data at a glance we have added box plots for each item (see table 4 of the manuscript). We thank Reviewer #2 for his/her helpful recommendation.

3. Have the authors made all data underlying the findings in their manuscript fully available?

Reviewer #1: Yes

Reviewer #2: No

OUR RESPONSE: The questionnaire in German and translated in English as well as anonymised data will be made available. Therefore, data availability statement will be changed as follows: “All relevant data are within the paper and its Supporting Information files.”. 

4. Is the manuscript presented in an intelligible fashion and written in standard English?

Reviewer #1: Yes

Reviewer #2: Yes

5. Review Comments to the Author

Reviewer #1: 

Table 3 - Title of third column could be more specific to show difference from column 2. e.g. proportion of those who had used elements of the campaign

OUR RESPONSE: Thank you for your advice. We added a more specific explanation in order to improve understanding. The new title of column 3 is as follows: “The proportion of those also being familiar with the campaign or who had used at least elements of a certain campaign.” (line number 245). We hope to clarify the difference between column 2 and 3.

Lines 248-250 - Sentence needs to express idea more clearly.

OUR RESPONSE: Thank you for your comment. Our description of the result may be misunderstood by readers. Therefore, we have decided to quote the item of the questionnaire verbatim. The sentence is now as follows: “A majority of GPs agreed to the statement that they already have “decided against therapeutic measures, even though the newly implemented lower threshold would have suggested treatment of some form" (Item 5.12, M = 4.5; SD = 1.6).” (line number 256-259). 

Limitations should include a comment on the low response rate; why this might be so e.g. response rates in electronic surveys by other authors; how this affects study; how this response rate might be improved in subsequent studies.

OUR RESPONSE: We appreciate the comment and have added the following information in the discussion part: “Response rate of 18% was limited in comparison to other surveys [36, 52]. Reasons might be the lack of incentive and/or a short recruitment time. Demographic variables were relatively close to the real distribution in the respective regions. GPs in our sample were slightly younger and proportion of men was higher. Nonetheless, findings should be interpreted with caution as generalization might be reduced. The voluntary participation and lack of financial compensation for participation can lead to a potential bias as it can be assumed that mostly GPs who are interested in medical overuse might have been more likely to respond to our questionnaire, skewing the sample.” (line number 367-375).

Reviewer #2: The article is on an important topic area—overuse of medical services.

The article is written clearly, and the tables are well designed. Overall, the authors followed a reasonable study design was followed. They used sound methods for developing and testing their questionnaire. The analysis has two deficits that can be amended before publication.

They used a registry of general practitioners (GPs) for sample selection. I was curious about why the authors limited their sample to GPs, as they noted that in Germany, unlike other countries, GPs do not serve as gatekeepers for access to medical specialists.

OUR RESPONSE: Thank you for your comment. As stated in the introduction section, the GP’s role as a gate-keeper is weakened. Nonetheless, in case of health problems also in Germany most people go to their GP first. It also seems that the role of GPs as gate-keepers will be strengthening as health insurances provide family doctor-centred health care where patients have to go to their GP first. Also, in rural areas and for vulnerable populations, e.g., for persons with chronic conditions, patients with multimorbidity and of higher age, the GP is the most important physician and often the coordinator of care. Therefore, we see the GP in a central role to prevent medical overuse and decided to specifically ask them about their views and opinions regarding overtreatment. 

The study has two substantial methodological issues that limit the utility of the results. The authors followed sound survey procedures, with the exception of survey fielding. They only allowed 18 days for GPs to return the survey and sent out a single reminder. A longer data collection period and more follow-up probably would have improved the response rate.

OUR RESPONSE: We thank for your comment. Short recruitment was due to our previous experience that response rate did not increase markedly after two invitations to participate. The following information have been added in the discussion part: Response rate of 18% was limited in comparison to other surveys [36, 52]. Reasons might be the lack of incentive and/or a short recruitment time. Demographic variables were relatively close to the real distribution in the respective regions. GPs in our sample were slightly younger and proportion of men was higher. Nonetheless, findings should be interpreted with caution as generalization might be reduced. The voluntary participation and lack of financial compensation for participation can lead to a potential bias as it can be assumed that mostly GPs who are interested in medical overuse might have been more likely to respond to our questionnaire, skewing the sample.” (line number 367-375).

Selection Bias. The study had a good sample plan, but only 18% of subjects returned complete questionnaires. An 18% response rate almost certainly entails some sort of selection bias. The authors mention that only respondents who are interested in the topic of medical overuse may have responded to the survey, although there is almost no way to test that. The authors report demographics for the physicians who returned the survey. I don’t know if they have access to demographics for the entire population of physicians in the geographic area, but if so, it would be helpful to compare the demographics of those who returned the survey with the entire population. While demographics would not tell much of a story about self-selection, it would be beneficial to know if there was selection by age or some other characteristic. In particular, since the authors found that younger doctors viewed medical overuse as more of a problem than older doctors, information on age of the sample compared with age of the population of GPs would have bolstered results.

OUR RESPONSE: Thank you for this recommendation. We added information regarding age and sex of the entire population of GPs within the respective area (line 217-220). Information is provided by the Bavarian Association of Statutory Health Insurance Physicians (https://www.kvb.de/fileadmin/kvb/dokumente/UeberUns/Versorgung/KVB-Versorgungsatlas_Hausaerzte.pdf Data from 2017 can be made available upon request).

Inefficient use of data. The questionnaire was comprised mainly of Likert-style questions on a scale of 1 to 6. The authors dichotomized responses into positive and negative. They provide no explanation about why they chose to do this, with the result that they lost information.

Suggestions for Revision

1. Compare the mean ages or age quartiles of survey respondents with the age of all GPs

2. Reanalyze the data without dichotomizing the Likert scales

OUR RESPONSE: We thank for this recommendation. As stated above, 

(1) Age of all GPs is now described in the result section, line 217-220

(2) Data of all Likert scales were reanalysed without dichotomizing. Descriptive analyses now comprise presentation of mean values, standard deviation and also box plots for each item are shown.

---

## [Decision Letter · Decision Letter 1]

19 Dec 2019

Is it really always only the others who are to blame? GP’s view on medical overuse. A questionnaire study.

PONE-D-19-23067R1

Dear Dr. Hueber,

We are pleased to inform you that your manuscript has been judged scientifically suitable for publication and will be formally accepted for publication once it complies with all outstanding technical requirements.

With kind regards,

Monika R. Asnani, DM, PhD

Academic Editor

PLOS ONE

Additional Editor Comments (optional):

Reviewers' comments:

Reviewer's Responses to Questions

**Comments to the Author**

1. If the authors have adequately addressed your comments raised in a previous round of review and you feel that this manuscript is now acceptable for publication, you may indicate that here to bypass the “Comments to the Author” section, enter your conflict of interest statement in the “Confidential to Editor” section, and submit your "Accept" recommendation.

Reviewer #1: All comments have been addressed

Reviewer #2: All comments have been addressed

2. Is the manuscript technically sound, and do the data support the conclusions?

Reviewer #1: Yes

Reviewer #2: Yes

3. Has the statistical analysis been performed appropriately and rigorously? 

Reviewer #1: Yes

Reviewer #2: Yes

4. Have the authors made all data underlying the findings in their manuscript fully available?

Reviewer #1: Yes

Reviewer #2: Yes

5. Is the manuscript presented in an intelligible fashion and written in standard English?

Reviewer #1: Yes

Reviewer #2: Yes

6. Review Comments to the Author

Reviewer #1: (No Response)

Reviewer #2: (No Response)

7. PLOS authors have the option to publish the peer review history of their article (what does this mean?). If published, this will include your full peer review and any attached files.

Reviewer #1: No

Reviewer #2: No

---

## [Editor Report · Acceptance letter]

23 Dec 2019

PONE-D-19-23067R1 

Is it really always only the others who are to blame? GP’s view on medical overuse. A questionnaire study. 

Dear Dr. Hueber:

I am pleased to inform you that your manuscript has been deemed suitable for publication in PLOS ONE. Congratulations! Your manuscript is now with our production department. 

With kind regards,

on behalf of

Dr. Monika R. Asnani 

Academic Editor

PLOS ONE